# Single AAV-mediated mutation replacement genome editing in limited number of photoreceptors restores vision in mice

Koji M. Nishiguchi [1,2,5]*, Kosuke Fujita[3,5], Fuyuki Miya[4], Shota Katayama[1] & Toru Nakazawa[1,2,3]*

Supplementing wildtype copies of functionally defective genes with adeno-associated virus (AAV) is a strategy being explored clinically for various retinal dystrophies. However, the low cargo limit of this vector allows its use in only a fraction of patients with mutations in relatively small pathogenic genes. To overcome this issue, we developed a single AAV platform that allows local replacement of a mutated sequence with its wildtype counterpart, based on combined CRISPR-Cas9 and micro-homology-mediated end-joining (MMEJ). In blind mice, the mutation replacement rescued approximately 10% of photoreceptors, resulting in an improvement in light sensitivity and an increase in visual acuity. These effects were comparable to restoration mediated by gene supplementation, which targets a greater number of photoreceptors. This strategy may be applied for the treatment of inherited disorders caused by mutations in larger genes, for which conventional gene supplementation therapy is not currently feasible.

---

[1] Department of Advanced Ophthalmic Medicine, Tohoku University Graduate School of Medicine, Sendai 980-8574, Japan. [2] Department of Ophthalmology, Tohoku University Graduate School of Medicine, Sendai 980-8574, Japan. [3] Department of Ophthalmic Imaging and Information Analytics, Tohoku University Graduate School of Medicine, Sendai 980-8574, Japan. [4] Department of Medical Science Mathematics, Medical Research Institute, Tokyo Medical and Dental University, Tokyo 113-8510, Japan. [5]These authors contributed equally: Koji M. Nishiguchi, Kosuke Fujita. *email: nishiguchi@oph.med.tohoku.ac.jp; ntoru@oph.med.tohoku.ac.jp

Delivery of wild-type copies of the defective gene (gene supplementation) in retinal dystrophy patients with loss of function mutations via adeno-associated virus (AAV) has shown promising therapeutic effects[1]. However, the stringent cargo limit of the vector (4700–5000 bps)[2] allows its application to only a fraction of the patients with mutations in relatively small pathogenic genes. For example, according to our recent genetic survey of Japanese patients with retinitis pigmentosa, the most frequent inherited retinal degeneration, more than 90% were shown to have mutations in larger genes untreatable by AAV-mediated gene supplementation[3]. Thus, vast majority of these patients require approaches other than AAV-mediated gene supplementation to treat their mutations, except for rare exceptions[4]. Recently, the CRISPR-Cas9-mediated allele knock-out genome editing strategy, based on non-homologous end joining (NHEJ) has been successfully applied to correct gain-of-function mutations via AAV[5–8]. One of the unique advantages of the genome editing approach is that it allows local treatment of the genome, such that the approach does not depend on the size of the target gene. However, genome editing for loss-of-function mutations in larger genes that require local replacement of the mutated sequence with a wildtype counterpart (mutation replacement) has not been successful in treatment of neuronal disorders primarily affecting neurons, due to its low editing efficiency[9–12]. This could be partly attributed to the requirement of two separate vectors for this approach, in which various components including Cas9, two guide RNAs (gRNAs) and U6 promoters, and DNA template and flanking homology arms all needs to be contained.

Recently, extremely small homology arms of ~20 bps (microhomology arms), relative to the conventional homology arms sized a few hundred bps or more, have been successfully applied to edit mammalian genome in vivo[13]. This system termed microhomology-mediated end joining (MMEJ) reportedly allows precise integration of a DNA donor in a desired genomic location[14]. In this study, we aim to develop a single AAV vector platform for mutation replacement genome editing using MMEJ. Through application of the platform in mouse models of retinal dystrophy, we show that a robust restoration of the visual function can be achieved, supported by an improved genome editing efficacy.

## Results

### Characterization of mutation replacement genome editing.
First, we generated mutants of preexisting retina-specific promoters and conducted in vivo AAV reporter assays (Supplementary Fig. 1a–e and Supplementary Table 1). The smallest promoter that maintained neural retina-specific transcription was a 93-bp mutant GRK1 promoter with reporter expression in 65.5% of the photoreceptors, including the cones (Supplementary Fig. 1c). This was used to drive SaCas9 (3.2 Kb) expression. We tested our single-AAV vector platform in $Gnat1^{IRD2/IRD2}$/$Pde6c^{cpfl1/cpfl1}$ mice; the Gnat1 and Pde6c defects in these mice cause blindness due to a functional lack of rods and cones[15], leaving behind only a residual cortical light response to brightest flashes[16] mediated by Gnat2[17]. This allows the clear observation of therapeutic effects. We used our platform to correct IRD2 mutations in Gnat1; these mutations constitute a homozygous 59-bp deletion in intron 4 (Supplementary Fig. 2a), preventing protein expression in the rods[18], which comprise ~75% of murine retinal cells[19]. Six gRNAs designed to flank the mutation were assessed with a T7 endonuclease 1 (T7E1) assay (Supplementary Fig. 2b, c and Supplementary Table 2). The gRNA pair (1 + 4) that excised the mutation most efficiently was selected.

The constructed prototype single-AAV vector (MMEJ vector; Fig. 1a and Supplementary Fig. 3a, f, g) that allows mutation replacement via MMEJ was then injected sub-retinally in 6M-old blind mice. Mutations of up to a few bp were designed in the gRNA target sites flanking the donor sequence to prevent repeated cleavage of the sites after successful mutation replacement. At this age, the rods show little sign of degeneration[20]. Six weeks later, histology showed scattered GNAT1-positive photoreceptors, indicating successful genome editing (Fig. 1b). Injection of a modified MMEJ vector that tagged SaCas9 expression with a fluorescent reporter (Supplementary Fig. 3b) showed GNAT1 immunoreactivity exclusively in the cells and retinal area with reporter expression (Fig. 1c), suggesting a causal relationship between SaCas9 and GNAT1 expression. Furthermore, histology showed no sign of accelerated cone degeneration as a side effect of the treatment, although we have no evidence that genome editing occurs in cones (Fig. 1d and Supplementary Fig. 1c). Next, we investigated the effects of Gnat1 mutation replacement on mRNA expression of related genes (Fig. 1e). The expression of Rho and Pde6b, both of which cooperate with Gnat1 to signal photo-transduction in rods[21], and of Rcvrn, a marker of both rods and cones, were not reduced in the eyes of untreated blind mice and remained unchanged after the treatment. However, the expression of the rod bipolar cell marker Pkcα, which had been reduced to 29.3% of its expression in the controls, nearly doubled to 50.0% following the treatment, indicating that the treated rods interacted with the downstream bipolar cells. Meanwhile, the absolute editing efficiency deduced from Gnat1 mRNA expression was ~12.7% (Fig. 1f). In contrast, when microhomology arms (MHAs) or gRNA target sites flanking the donor sequence were removed from the prototype MMEJ vector (Supplementary Fig. 3c, d), the efficiency was dramatically reduced, consistent with mutation replacement mediated by MMEJ. Furthermore, testing with a 6-Hz flicker electroretinogram (ERG), which reflects the number of functional photoreceptors, revealed responses averaging 11.2% of that in the control mice (Fig. 1g). The effect was severely diminished after the intravitreal injection of LAP4, a glutamate analog that blocks synaptic transmission between the photoreceptors and ON-bipolar cells[20]. This is consistent with functional connection of the treated rods with downstream neural circuits. The result was further corroborated by a single-flash ERG paradigm: mice pretreated with MMEJ vector and then injected with LAP4 showed reduced b-waves generated by the ON bipolar cells including the rod bipolar cells, and preserved a-waves driven by rods (Fig. 1h). Again, the modified vectors without MHAs or gRNA target sites, showed no discernable response in either ERG protocol, supporting the specific role of MMEJ in mutation replacement. These results were consistent with ~10% success in mutation replacement via MMEJ in the rods and functional integration of the treated cells into the retinal circuitry.

### On- and off-target analysis.
Next, we carried out PCR-based sequencing analyses of the on-target site in vitro (Supplementary Fig. 4) and in vivo (Fig. 2). The in vitro analysis showed a 10.3% success rate after MMEJ mutation replacement, higher than the rate of 3.8% with a different mutation replacement strategy (homology-independent targeted integration, HITI; Supplementary Figs. 3e–g and 4a, b)[9]. Similarly, the success rate of in vivo mutation replacement in the genome-edited rods was 11.1% and 4.5% for the MMEJ and HITI approaches, respectively, at 1 M post-treatment (Fig. 2a). Gross estimate of absolute successful editing rate in the rods, uncorrected and partially corrected also for the sensitivity of the sequence analysis, was 4.7% and 9.1% for the MMEJ approach (Fig. 2b–d). In both the in vitro and in vivo

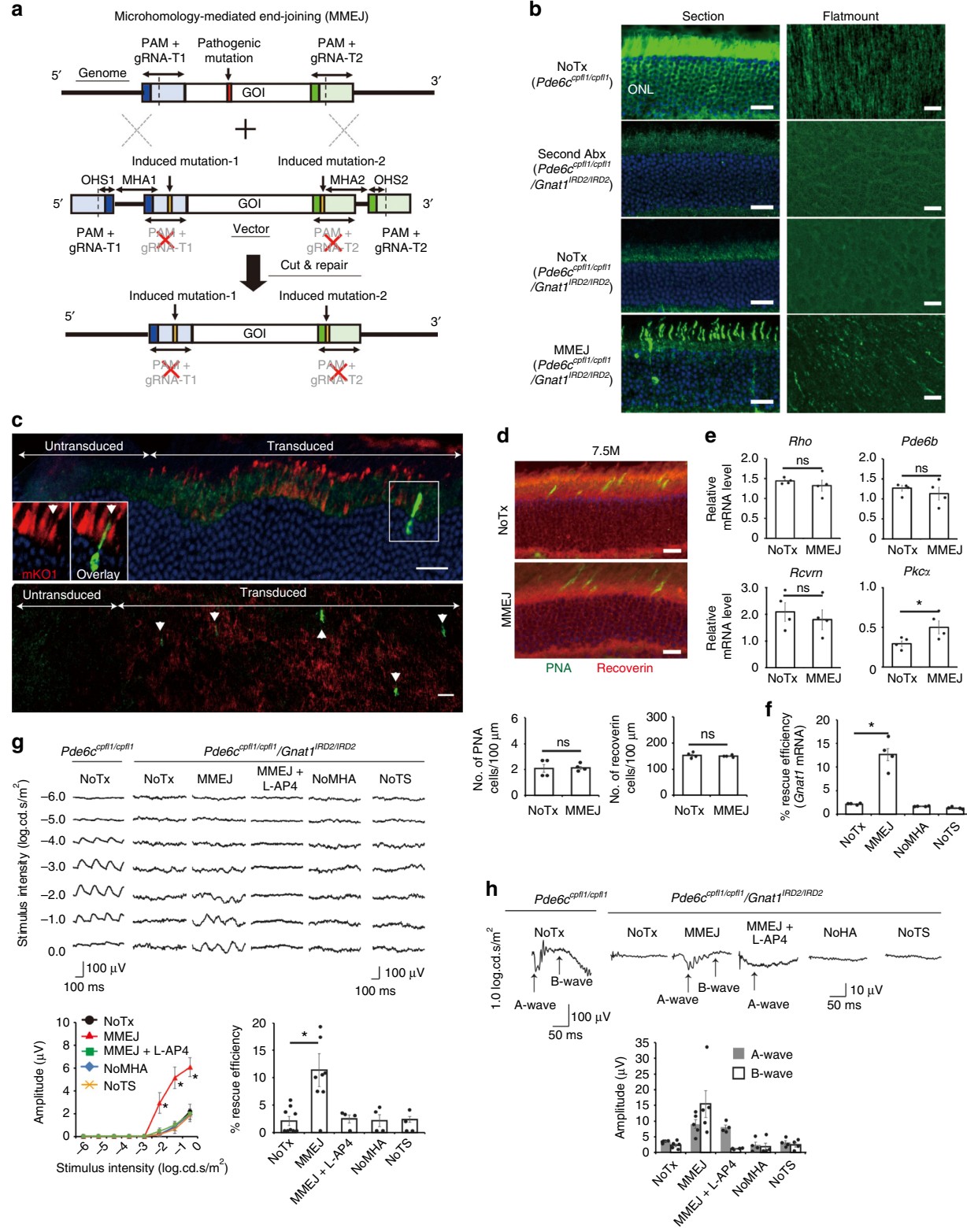

analyses, MMEJ vectors without MHAs or gRNA target sites did not result in any successful mutation replacements. Meanwhile, the major editing outcome was deletion caused by a simple excision of the *IRD2* mutation for both in vitro and in vivo analyses. Unplanned in vivo on-target integrations of the AAV genome were present, but at a lower rate than deletions. Extended in vivo on-target site sequencing and mRNA analysis (Fig. 2a–f) conducted 3 M post-treatment revealed a similar absolute success

rate (corrected editing rate of 11.0%) accompanied by the sustained or slightly reduced expression of SaCas9 mRNA and gRNAs (Fig. 2g, h), demonstrating the stability of the platform. The result also indicates that the treatment effect nearly plateaus by 1 M. Although accurate estimation by PCR-based sequencing is difficult, the results support the stable ~10% absolute editing efficiency at the genome level in the rods with MMEJ-mediated mutation replacement.

**Fig. 1 In vivo characterization of mutation replacement genome editing. a** Illustration of MMEJ-mediated mutation replacement. Genome of interest (GOI) with and without the mutation are excised at the flanking gRNA target sites (gRNA-T1 and -T2; dotted line) from mouse genome and AAV vector, respectively, by SaCas9 and two gRNAs. GOI without mutation is inserted into the genome using microhomology arms (MHA), thereby correcting the mutation. **b** GNAT1 staining. GNAT1-positive photoreceptors (arrowhead) were observed (section, left; flatmount, right). **c** Co-localization of Kusabira Orange (mKO1, red) probing SaCas9 expression and GNAT immunopositivity (inset, green). Scattered GNAT-positive cells were observed only in the area transduced with mKO1 (section, top; flatmount, bottom). Note, oversized reporter vector (5201 bp) drastically reduced editing efficiency. $N = 4$ **d** PNA and recoverin staining with quantification ($N = 4$). **e** RT-PCR of *Rho*, *Pde6b*, *Rcvrn*, and *Pkcα* (relative to *Pde6c*$^{cpfl1/cpfl1}$ mice; $N = 4$ for all). **f** Rescue efficiency by RT-PCR of *Gnat1* (relative to *Pde6c*$^{cpfl1/cpfl1}$ mice; $N = 4$ for all). **g** 6-Hz flicker ERGs. $N = 9$, 9, 4, 4, and 4 for No treatment (NoTx), MMEJ, MMEJ + L-AP4, NoMHA, NoTS, respectively. In MMEJ + L-AP4, MMEJ vector and L-AP4 were sequentially injected. Amplitudes ($-1.0$ log.cd.s.m$^{-2}$) relative to those of *Pde6c*$^{cpfl1/cpfl1}$ mice indicate %rescue efficiency (bottom right). **h**. Single flash ERGs. The same group of mice used in **g**. Scale bar: 20 μm; Data represent mean ± S.E.M.; *$P < 0.05$ (Student's *t*-test); ns, not significant; PAM, protospacer adjacent motif, OHS, over-hanging sequence; Abx, antibodies; NoTS, no gRNA target sites. Source data are provided as a Source Data file.

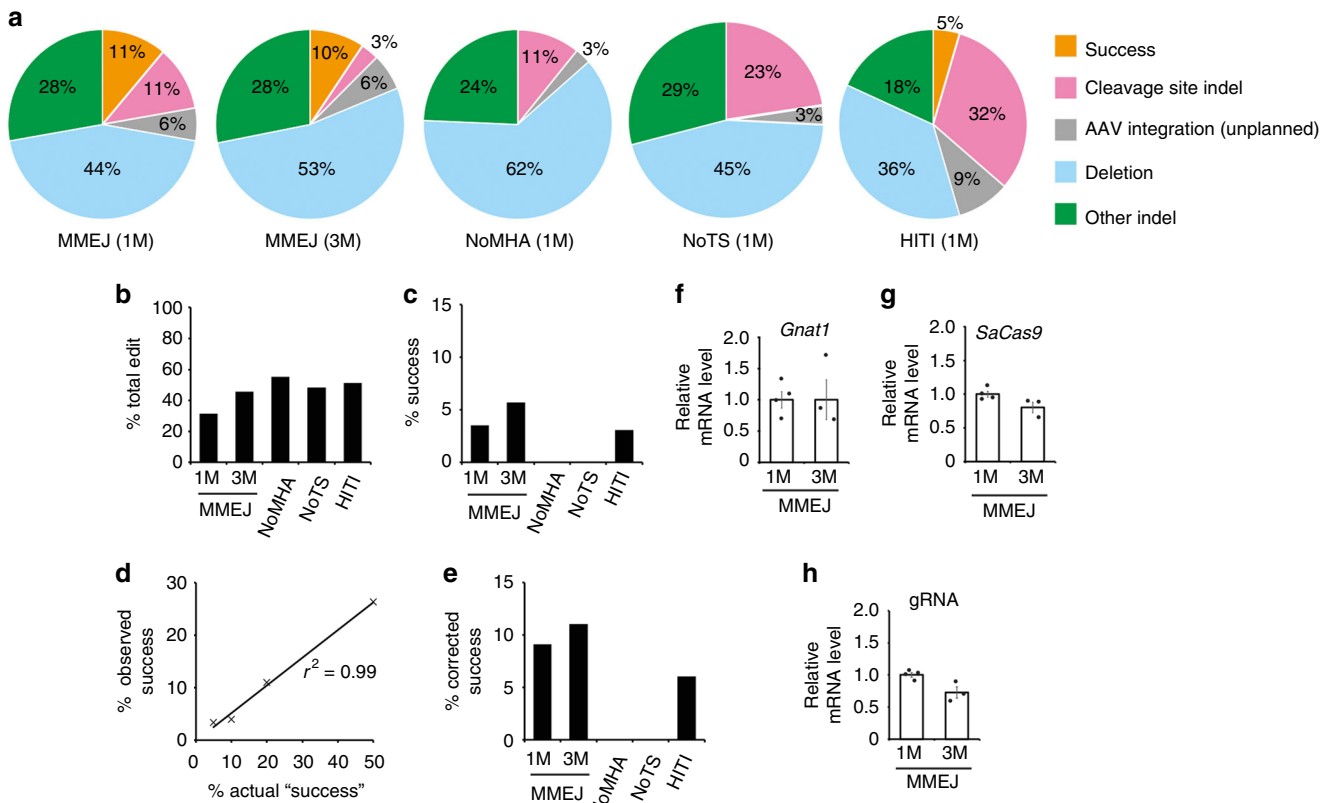

**Fig. 2 In vivo assessment of the on-target site. a** Breakup of sequencing results of the on-target site in the genome-edited clones amplified from the retina collected 1 M or 3 M post-injection. MMEJ, NoMHA, and NoTS represents injection of protype MMEJ vector, MMEJ vector without microhomology arms, and MMEJ vector without gRNA target sites, respectively. HITI represents homology-independent targeted integration. See Supplementary Fig. 3 for vector map. Total clones sequenced were 57, 70, 67, 64 and 86 for MMEJ (1 M), MMEJ (3 M), NoMHA (1 M), NoTS (1 M), and HITI (1 M), respectively. Success indicates successful mutation replacement. Cleavage site indel represents indels in either gRNA cleavage site without replacement of *IRD2* mutation (see Online Methods for detail). Note, co-existing mutation replacement and cleavage site indels, which is expected to occur as a consequence of repeated cleavage at the gRNA site was not observed. **b** Total editing rate. Percentage of clones that showed any sign of genome editing among the total clones analyzed. **c** Absolute success rate in the rods, assuming the cells comprise 75% of retinal neurons. **d** Estimation of detection efficiency of Success allele by subcloning and PCR. Observed % Success (vertical axis) were obtained by distinguishing the identity of the clones derived from PCR products amplified from mixture of Success and unedited mutant DNA templates at various ratio (horizontal axis). Total clones sequenced in this experiment were 61, 52, 64 and 61 for 5%, 10%, 20%, and 50% mixtures (rate of Success DNA versus total DNA), respectively. Intercept = $-0.154$, slope = 0.528, $r^2 = 0.99$. **e** Estimated absolute success rate in the rods corrected for by the detection efficiency of Success alleles relative to unedited mutant allele, assuming 75% of retinal neurons are the rods. **f** RT-PCR of *Gnat1* at 1 M ($N = 4$) and 3 M ($N = 3$). **g** RT-PCR of *SaCas9* at 1 M ($N = 4$) and 3 M ($N = 3$) post-injection. **h**. RT-PCR of gRNA at 1 M ($N = 4$), and 3 M ($N = 3$) post-injection. Data represent the mean ± S.E.M. Source data are provided as a Source Data file.

Then, off-target analysis was performed with a T7E1 assay and PCR-based sequencing of 14 predicted sites (7 for each gRNA, Supplementary Table 2). These showed no mutation events in retinas collected 1 M after MMEJ vector injection (Supplementary Fig. 5). In these sites, whole-genome sequencing of 4 retinas of 4 mice collected 1 M post-injection (average read depth of 158 per

base) and an additional three retinas of three mice collected 4 M post-injection (average read depth of 126 per base) revealed no off-target events (Supplementary Table 3). In addition, we listed up additional 59 potential off-target sites in an unbiased manner, by selecting all variants in whole-genome sequence data that were present in 3 independent samples collected after the therapeutic

transfection of murine Neuro2A cells, but were not present in a single sample collected before transfection (Supplementary Table. 3). No indels were observed in these sites using the whole-genome sequence data from the seven retinas also used for the on-target analysis (average read depth of 219 per site). Furthermore, there was no evidence of AAV integration into the mouse genome outside of the on-target site. Together, these results indicate that off-target indel formation was rare, if it occurred at all.

**Visual restoration by mutation replacement genome editing.** Next, we investigated the therapeutic effects of MMEJ-mediated mutation replacement. Light sensitivity in the visual cortex was assessed with flash visually evoked potentials (fVEPs). Surprisingly, cortical responses contralateral to the treated eye revealed a ~10,000-fold (range: 1000–100,000-fold) improvement in light sensitivity, equivalent to gene supplementation in ~70% of the photoreceptors (Fig. 3a, Supplementary Figs. 1f, 3h, and 6a) with greater ERG rescue (Supplementary Fig. 6b, c)[16]. Changes in light-induced behavior (fear conditioning, Fig. 3b) also reflected this improvement. Furthermore, cortical responses to phase-reversal gratings of various spatial resolutions, i.e., the pattern VEP (pVEP), showed larger amplitudes post-treatment (Fig. 3c). The threshold of spatial resolution of vision (i.e., visual acuity), determined by measuring the optokinetic response (OKR), was restored in the treated mice to 59.1% of the control mice, also similar to the effect of gene supplementation (Fig. 3d). Taken together, MMEJ-mediated *Gnat1* mutation replacement allowed substantial improvement of light sensitivity and visual acuity, comparable to the effects delivered by gene supplementation.

We also used MMEJ-mediated mutation replacement to treat 2M-old *Gnat1*[IRD2/IRD2] mice, which retain cone function and serve as a model of human retinal dystrophy[22]. In the early course of the disease, patients suffer from severe loss of light sensitivity with preserved visual acuity[22]. A histological analysis showed scattered GNAT1-postive photoreceptors in the treated mice (Fig. 4a). RT-PCR measurement indicated that absolute genome editing efficiency was 7.2% (Supplementary Fig. 7a). The fVEP analysis showed a ~1000-fold increase in light sensitivity (Fig. 4b). This was confirmed behaviorally in a fear conditioning experiment (Fig. 4c). However, the improvement in retinal function could not be isolated from preexisting cone function by ERG testing, and visual acuity remained unchanged in pVEP and OKR testing (Supplementary Fig. 7b–d). These results show that the therapeutic effects of our platform extended to an animal model of human disease.

## Discussion

This study shows that mutation replacement genome editing with a single AAV vector can achieve striking improvements in light sensitivity and visual acuity comparable to that of gene supplementation[16]. The results showed that the gene supplementation can treat by far a larger number of retinal neurons compared to the mutation replacement genome editing, resulting in substantially larger ERG responses directly proportional to the increased number of light-responsive photoreceptors in the former. However, the light sensitivity as defined by dimmest recognizable light stimulus and visual acuity was not very different between the two treatment approaches (Fig. 3a right lower panel). This is because thresholds of these visual perceptions reflect functional integrity of defined number of photoreceptors rather the total number of treated retinal neurons.

This therapeutic platform renders a major step forward from the dual vector-based mutation replacements, which generally yield an absolute editing efficiency of less than ~5%[10–12] in post-

mitotic cells, including 4.5% efficiency at the level of mRNA in the retinal pigment epithelium in a rat model of retinal dystrophy[9], compared to the efficiency of up to ~10% shown here by genomic, mRNA, and functional analysis. This paves the way for treating loss-of-function mutations in larger genes, for which conventional gene supplementation therapy or NHEJ-based genome editing strategies are not generally feasible.

## Methods

**Animals.** *Pde6c*[cpfl1/cpfl1]*Gnat1*[IRD2/IRD2] mice (i.e., blind mice) were derived from *Gnat1*[IRD2/IRD2] mice (Takeda, Japan)[18], which are rod-defective, and *Pde6c*[cpfl1/cpfl1] mice (Jackson Laboratory, Bar Harbor, ME)[23], which are cone-defective. The phenotype of these mice has been previously studied and reported[16]. For the in vivo reporter assay, an AAV vector ($1 \times 10^{12}$ gc mL$^{-1}$) was injected (1.5 μL per injection) into the ventral subretinal space of 3-month-old C57BL/6 J mice (Japan SLC Inc., Hamamatsu, Japan)[24]. For mutation placement genome editing, the AAV vector ($1 \times 10^{12}$ gc mL$^{-1}$) was injected (1.5 μL per injection) into the dorsal and ventral subretinal space of *Pde6c*[cpfl1/cpfl1]*Gnat1*[IRD2/IRD2] mice (6-month-old) and *Gnat1*[IRD2/IRD2] mice (2-month-old). Control animals comprised age-matched *Pde6c* [cpfl1/cpfl1] mice or C57BL/6 J mice (Japan SLC Inc., Hamamatsu, Japan). The surgical procedures were performed after intraperitoneal administration of a mixture of ketamine (37.5 mg kg$^{-1}$) and medetomidine (0.63 mg kg$^{-1}$). The medetomidine was reversed by intraperitoneal administration of atipamezole (1.25 mg kg$^{-1}$) after the surgery. Sample sizes were calculated using an on-line sample size calculator (https://www.stat.ubc.ca/) adopting a two-sided alpha-level of 0.05, 80% power. The parameters included the means and standard deviation predicted from a previous study we conducted with a similar experimental approach to evaluate effects of AAV-mediated gene supplementation therapy on a group of mice that had a similar genetic background[16]. Rarely, the sample size was limited by the availability of mice. The mice were handled in accordance with the ARVO Statement On the Use of Animals in Ophthalmic and Vision Research and the Tohoku University guidelines for the care and use of animals. All experimental procedures were conducted after approval by the relevant committee for animal experiments at Tohoku University Graduate School of Medicine.

**Miniaturization of photoreceptor-specific promoter.** Various known small promoters (Supplementary Table 1) were tested before deletion mutant promoters were synthesized by modifying the *RCV* promoter[25] or *GRK1* promoter[26] (Supplementary Fig. 1c, Thermo Fisher Scientific, Waltham, MA; Eurofins Genomics, Tokyo, Japan). They were each sub-cloned into a pAAV-MCS Promoterless Expression Vector (Cell Biolabs Inc., San Diego, CA) containing an enhanced green fluorescent protein (EGFP) gene as a reporter[24,27]. AAV2/8 containing the reporter constructs were generated and purified following the method described below. Each virus was injected into two eyes of C57BL/6J mice. The eyes were collected 1 week after the injection and were processed for histological assessment as described below.

**Selection of gRNAs.** Three gRNAs each were designed using Cas-Designer (http://www.rgenome.net/cas-designer/) on both sides of the mutation in the intron 4 and exon 4 of *Gnat1*, as displayed in Supplementary Fig. 2 (Assembly: GRCm38/mm10). Oligos for each gRNA were subcloned with pX601 vector (gRNA expression plasmid, addgene #61591). Neuro2a cells (Cell Resource Center for Biomedical Research, Tohoku University, Sendai, Japan) were transfected with an gRNA expression plasmid using lipofectamine 3000 transfection reagent (Thermo Fisher Scientific). Genomic DNA was extracted 72 h post-transfection using a DNA extraction kit (QIAamp DNA Mini kit; Qiagen, Hilden, Germany). This was subjected to a T7E1 assay following the manufacturer's instructions (New England Biolabs, Ipswich, MA). In brief, genomic fragments containing the gRNA target site were amplified with PCR and purified using NucleoSpin Gel and a PCR Clean-up kit (Macherey-Nagel, Düren, Germany). Then 200 ng of each of the PCR products derived from the transfected and non-transfected cells were denatured at 95 °C for 5 min and reannealed, then digested with T7E1 for 30 min at 37 °C, followed by electrophoresis in 2% agarose gel. After measuring the density of the bands with ImageJ, %indels was calculated following the formula: $100 \times (1 - (1 - \text{cleaved band intensity/total band intensities})1/2)$[28]. The sequences of all the PCR primers used in this study are presented in Supplementary Table 2.

**Construction and purification of plasmid and AAV vectors.** The all-in-one CRISPR/SaCas9 plasmid (pX601) for mutation replacement genome editing was assembled as shown in Supplementary Fig. 1. The 93-bp *GRK1* promoter (*GRK1-93*) was used to drive SaCas9 (Ac. No. CCK74173.1; from pX601) expression. Oligonucleotides for the donor template, which comprised the flanking micro-homology arms, gRNA target sites and the donor sequence, were synthesized and inserted into the vector using a DNA ligation kit (Clontech, Mountain View, CA). To avoid repeated cleavage after successful mutation replacement, mutations were introduced in the flanking gRNA target sites. The 4 bp mutation in the 5′ gRNA-1 target site inside exon 4 was selected using codon optimization tool GENEisu

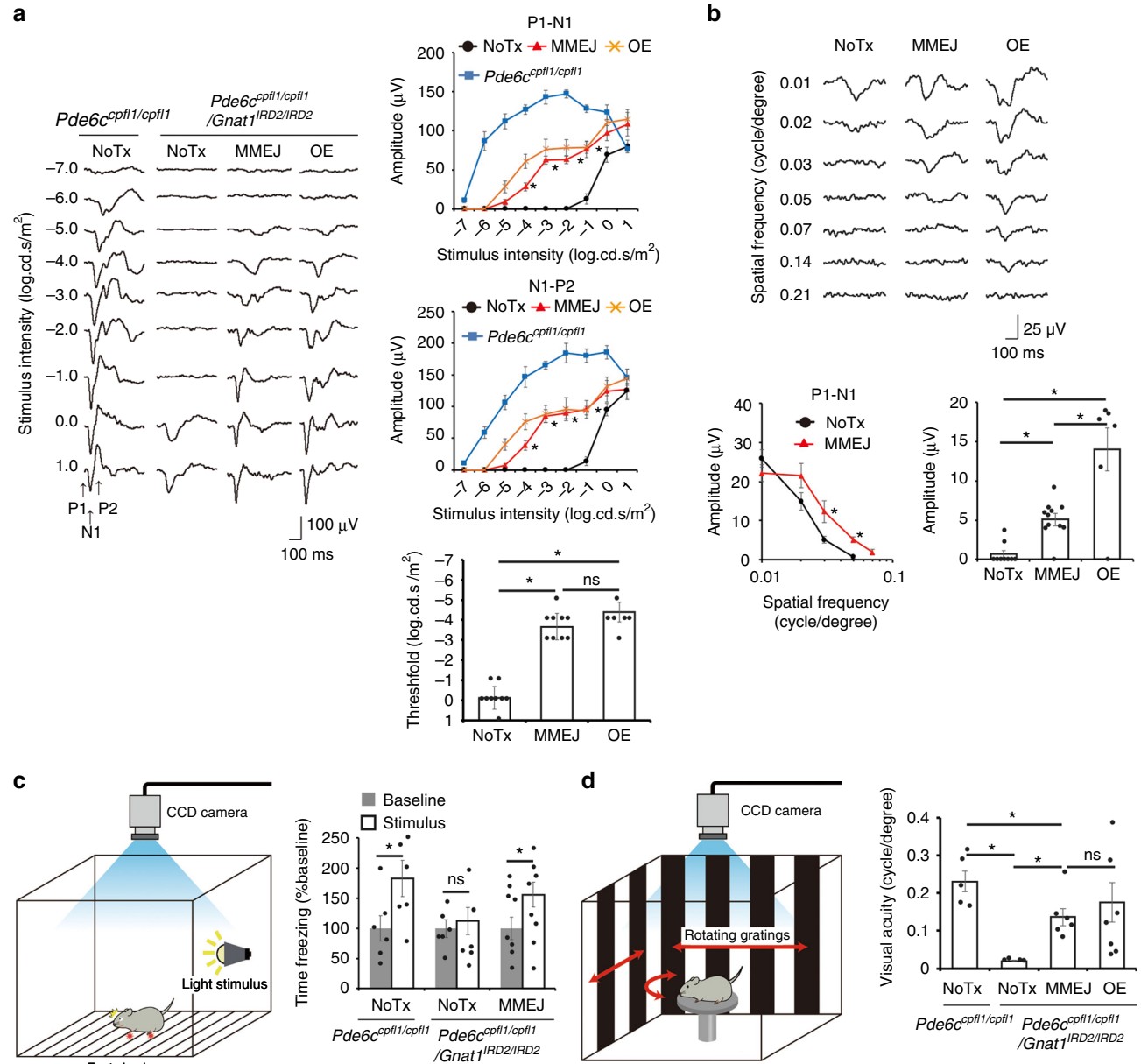

**Fig. 3 Visual restoration by in vivo mutation-replacement genome editing.** Flash visually evoked potentials (fVEP) of the visual cortex contralateral to the eyes in response to flashes of various intensities. MMEJ indicates eyes treated with *Gnat1* mutation replacement ($N = 9$) and OE (over-expression) indicates those with *Gnat1* gene supplementation ($N = 6$), both delivered by single AAV. NoTx refers to untreated eyes ($N = 9$). Control *Pde6c^{cpfl1/cpfl1}* mice ($N = 5$). Note, light sensitivity as defined in the Methods was increased by ~4 log unit after MMEJ-mediated genome editing, which was not significantly different to the effect mediated by OE (right lower panel). **b** Pattern VEPs. $N = 11$, 10, and 6 for MMEJ, untreated and OE, respectively. **c** Fear conditioning test. Freezing time before (Baseline) and during (Stimulus) presentation of fear-conditioned light cue from MMEJ treated ($N = 9$) and untreated ($N = 6$) mice. **d** Optokinetic response. Note threshold of spatial resolution of vision (visual acuity) was not different in the MMEJ and OE. $N = 10$, 7, and 4 for MMEJ, OE, and NoTx, respectively. Control *Pde6c^{cpfl1/cpfl1}* mice ($N = 6$). Data represent the mean ± S.E.M.; *$P < 0.05$ (**a**, **b**, **d**, ANOVAs followed by Tukey's post hoc test; **c** Student's *t*-test); nd, non-detectable; ns, not significant. Source data are provided as a Source Data file.

(http://www.geneius.de/GENEius/). For selecting 1 bp mutation in the 3′ gRNA-4 target site, the corresponding genomic sequences from *Mus musculus, Mus Caloli, Mus phari* and *Rattus norvegicus* were aligned by ClustalW (https://clustalw.ddbj.nig.ac.jp/). The sequences were perfectly conserved except for a single variant in *Rattus norvegicus*, which was chosen for inducing mutation. Mutations at both target sites were confirmed with off-target site analysis tool CRISPOR (http://crispor.tefor.net/) to yield lowest probability of cleavage (Cutting frequency determination score of 0.00). For labeling of SaCas9 expression, 2A peptide and mKO1 (monomeric Kusabira-Orange 1) red fluorescence protein cDNA (MBL, Nagoya, Japan) were inserted downstream of SaCas9 into the vector using a NEBuilder HiFi assembly kit (New England Biolabs). For construction of plasmid used for cell sorting, MMEJ vector was modified so that 2A peptide and EGFP

cDNA (Clontech) were inserted downstream of the SaCas9 driven by *CMV* promoter replaced for the *GRK1-93* promotor. The NoMHA (no mirohomology arm), NoTS (no target site) and HITI (homology-independent targeted integration) plasmids was assembled as shown in Supplementary Fig. 1. Each fragment was synthesized and inserted into the vector using the same regents described above. We also constructed a plasmid vector for gene supplementation of *GNAT1*[16] (shown in Supplementary Fig. 1h). In brief, full length *GNAT1* cDNA (KIEE3139; Promega Corp., Madison, WI) was subcloned downstream of the ubiquitous *CMV* promoter into the AAV-MCS vector (Cell Biolabs Inc).

Then, the constructed plasmid vectors were used for in vitro assays or assembled into AAV2/8. In brief, each vector was co-transfected with AAV2rep/AAV8cap vector (pdp8; Plasmid Factory, Bielefeld, Germany) in HEK293T cells

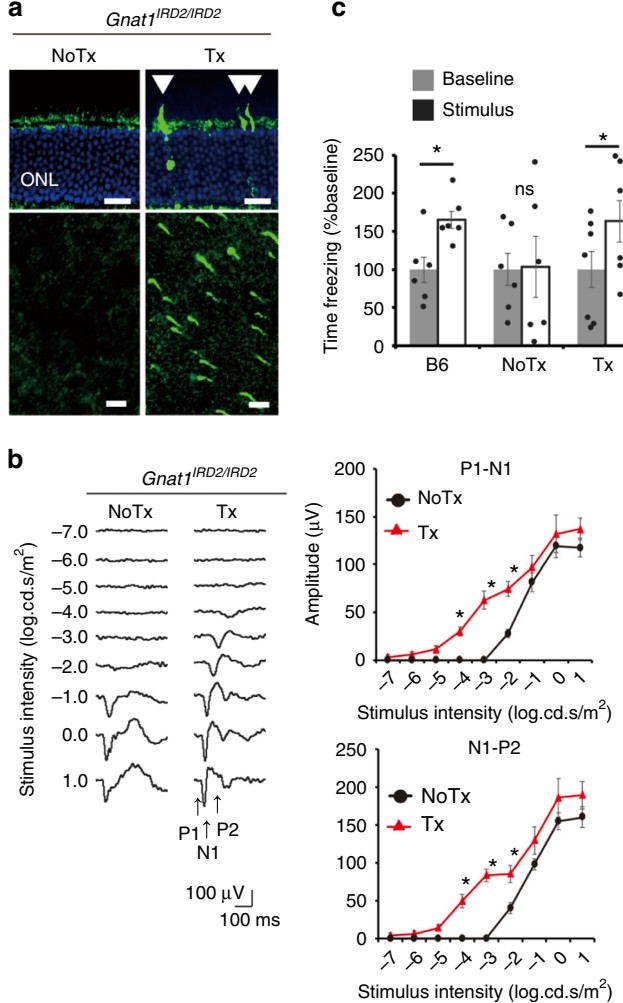

**Fig. 4 In vivo mutation replacement genome editing in a mouse model of retinal degeneration. a** GNAT1-positive photoreceptors (arrowhead) following treatment of Gnat1$^{IRD2/IRD2}$ mice shown in a retinal section (top) and a flatmount (bottom). Scale bar: 20 μm. **b** fVEPs recorded from contralateral visual cortices in treated and untreated eyes of the same mice. ($N = 7$). **c** Fear conditioning test, showing freezing time before (Baseline) and during (Stimulus) presentation of fear-conditioned light cue. Treated (Tx, $N = 7$) and untreated (NoTx, $N = 6$) Gnat1$^{IRD2/IRD2}$ mice and CL57B6 mice (B6, $N = 6$). Data represent the mean ± S.E.M.; *$P < 0.05$ (Student's t-test); ONL, outer nuclear layer. ns, not significant. Source data are provided as a Source Data file.

(Thermo Fisher Scientific) using PEI (Polysciences Inc, Warrington, PA). AAV particles was extracted in PBS and purified with an AKTA prime plus chromatography system (GE Healthcare, Chicago, IL) on an AVB Separose HP column (GE Healthcare)[24,27].

**RT-PCR**. Total RNA was purified from the mouse retinas using the miRNeasy plus mini kit (Qiagen) and reverse-transcribed with SuperScript III (Thermo Fisher Scientific). qRT-PCR was performed with an initial denaturation step at 95 °C for 20 s, followed by 40 cycles at 95 °C for 3 s and 60 °C for 20 s (7500 Fast Real-Time PCR System; Thermo Fisher Scientific). Taqman probes for Gnat1 (Mm01229120; Thermo Fisher Scientific), Gapdh (Mm99999915; Thermo Fisher Scientific), Pde6b (Mm00476679; Thermo Fisher Scientific), Pkca (Mm00440858; Thermo Fisher Scientific), Rho (Mm01184405; Thermo Fisher Scientific), Rcvrn (Mm00501325; Thermo Fisher Scientific) and SaCas9 (AP2XCCY; Thermo Fisher Scientific) were used. For SaCas9 and gRNA scaffold, primers and probes[4] were shown in Supplementary Table 2. Each mRNA expression was determined by plotting CT values on the standard curve generated by serially diluting the control sample (C57BL/6J mice retinal cDNA or AAV injected retinal cDNA).

**Western blot**. The eyes were harvested 6 weeks (1.5 M) after the AAV injection. The retina and RPE/choroid complex were then collected separately and placed on ice. The tissues were dissolved in RIPA buffer and the total protein concentration was measured with a Pierce BCA protein assay kit (Thermo Fisher Scientific). Proteins (15 μg each) were separated based on their molecular weight with SDS–PAGE on 10% Mini-PROTEAN gels (Bio-Rad, Hercules, CA) and then transferred to PVDF membranes (Millipore, Billerica, MA). Membranes were blocked with 5% skim milk for 1 hr, incubated with rabbit anti-GFP antibody (#598, 1/2500; MBL) for 1 hr, and then incubated with horseradish peroxidase (HRP)-conjugated anti-rabbit IgG antibodies (A0545, 1/2000; Sigma-Aldrich) for 1 hr. The immunogenic signal was detected with ECL prime (GE Healthcare). The membrane was stripped and incubated with anti-beta-actin (F5316, 1/2000; Sigma-Aldrich, St. Louis, MO), incubated with HRP-conjugated anti-mouse antibodies (#31430, 1/2000; Thermo Fisher Scientific), and detected with ECL prime.

**Immunohistochemistry**. Immunohistochemistry was performed as follows[27]. Six weeks (1.5 M) after the AAV injection, the eyes were fixed in 4% paraformaldehyde. They were then either embedded in OCT compound and sectioned using a cryostat to generate retinal sections, or the RPE/choroid complex was separated from the retina and then flattened by creating four incisions from the periphery to the optic nerve, thereby resulting in a clover-shaped RPE/choroid flatmount. The retinal sections were blocked with 5% normal goat serum for 30 min, incubated with mouse anti-mKO1 monoclonal antibodies (M104-3M, 1/200; MBL), rabbit anti-M-opsin antibodies (AB5405, 1/1000; Millipore) or rabbit anti-recoverin (AB5585, 1/5000; Millipore) and Alexa Fluor 488 conjugated PNA (10 μg/mL, Thermo Fisher scientific) for 1 h, incubated with Alexa Fluo 568-conjugated anti-mouse IgG antibodies (1/500; Thermo Fisher Scientific) or Alexa Fluo 568-conjugated anti-rabbit IgG antibodies (1/500; Thermo Fisher Scientific) for for 1 h, and DAPI (Vector Labs, Burlingame CA) for additional 45 min. For the reporter assay, RPE/choroid flat mounts were stained with only DAPI for 45 min before imaging. For the analysis of GNAT1 expression, immunohistochemistry was carried out using a TSA Plus Fluorescein System (Perkin-Elmer, Waltham, MA, USA) following the manufacturer's instructions. In brief, the sections were blocked with 1% skim milk for 1 h, incubated with rabbit anti-GNAT1 antibodies (ab74059, 1/200; Abcam, Cambridge, UK) for 1 h, incubated with HRP-conjugated anti-rabbit IgG antibodies (A0545, 1/2000; Sigma-Aldrich) for 1 h, and then stained with TSA reagent and/or DAPI for an additional 45 min. The retinal flat mounts were stained as follows[29]. The isolated eyes were fixed in 4% paraformaldehyde. Then they were treated with three freeze/thaw cycles. For the analysis of Gnat1 expression, the retinas were blocked with 1% skim milk for 1 hr, incubated with rabbit anti-GNAT1 antibodies (ab74059, 1/200; Abcam, Cambridge, UK) overnight, incubated with HRP-conjugated anti-rabbit IgG antibodies (A0545, 1/2000; Sigma-Aldrich) for 1 h, and then stained with TSA reagent (Perkin-Elmer). For the analysis of mKO1 expression, the retinas were blocked with 5% normal goat serum for 1 h, incubated with mouse anti-mKO1 monoclonal antibodies (M104-3M, 1/200; MBL) overnight, incubated with Alexa Fluo 568-conjugated anti-mouse IgG antibodies (1/500; Thermo Fisher Scientific) for 1 h. Images were acquired on a Zeiss LSM780 confocal microscope (Carl Zeiss, Jena, Germany).

**Electrophysiological assessment**. Basic equipment and techniques for ERG and fVEP recordings were recorded using an PuREC (Mayo Corp., Inazawa, Japan) acquisition system, and an LED stimulator (LS-100; Mayo)[30]. For scotopic 6-Hz flicker ERGs[31], we used flash intensities at seven steps, ranging from −6.0 to 0 log cd s m$^{-2}$, separated by 1.0 log units. For each step, after 10 s of adaptation, 500 msec sweeps were recorded 50 times and averaged.

For standard single flash ERGs[16], we used flash intensities comprising 10 steps, ranging from −7.0 to 2.0 log cd s m$^{-2}$, separated by 1.0 log units (Supplementary Fig. 6c). Then, the standard protocol was optimized for an accurate estimation of the small effect of MMEJ-mediated mutation replacement in Pde6c$^{cpfl1/cpfl1}$Gnat1$^{IRD2/IRD2}$ mice, in which a fixed flash (1.0 log cd s m$^{-2}$) separated by 10 s intervals with increased averaging of 50 times (compared to 2 times in the standard protocol for this flash intensity) were applied (Fig. 1h). For assessing synaptic transmission between photoreceptor and ON-bipolar cells, group III mGlu agonist L-AP4 (L-2-amino-4-phosphonobutyric acid, ab12002, 50 mM; Abcam) was injected into the vitreous of the mice at 3 W after treatment with MMEJ vector and ERGs were recorded before (to ensure successful mutation replacement) and 20 h after injection[20].

Surgical implantation of the VEP electrodes were placed in primary visual cortex[30,32] 5 weeks post injection, and recording was performed a week later. For recording fVEPs, we used flash intensities at nine steps, ranging from −7.0 to 1.0 log cd s m$^{-2}$, separated by 1.0 log units. The light sensitivity of the visual cortex was determined by identifying the dimmest light condition that yielded an amplitude of the negative trough (P1-N1) or a positive peak (N1-P2) over 25 μV during fVEP recording. To record pVEPs, we used black (3 cd m$^{-2}$) and white (159 cd m$^{-2}$) vertical stripes of equal width (average luminance: 81 cd m$^{-2}$) with different spatial resolutions (0.42, 0.35, 0.28, 0.21, 0.14, 0.07, 0.05, 0.03, 0.02, and 0.01 cycles per degree for the Gnat1$^{IRD2/IRD}$ mice, and 0.21, 0.14, 0.07, 0.05, 0.03, 0.02 and 0.01 cycles per degree for the Gnat1$^{IRD2/IRD2}$/Pde6c$^{cpfl1/cpfl1}$ mice)[32]. The amplitudes for the negative trough (P1-N1) and positive peak (N1-P2) were plotted vertically as a function of the log spatial resolution of the stimulus (horizontally).

**Behavioral tests**. Fear conditioning was performed 3 weeks after the AAV treatment[20]. In the training session, each mouse was placed in a shock chamber with a stainless-steel grid floor (21.5 cm width × 20.5 cm depth × 30 cm height box; Ohara Medical Industry, Tokyo, Japan), located inside a sound attenuating box, and left for 2 mins to adapt to the environment. Then, the mouse was exposed to an LED light cue (535 nm, 0.015 cd m$^{-2}$, 2.0 Hz, 5.0 s) controlled via a stimulus controller (FZ-LU, Ohara Medical Industry) that co-terminated with a 0.8-mA foot shock (2.0-s duration). This was repeated five times at pseudorandomized intervals (70–140 s) before returning the mouse to the housing cage. In the testing session, which took place 24 h after the training session, the mice were returned to the same chamber to test for visually cued memory recall. In order to change the environmental context from the training session, a white floor and curved wall made of thin plastic were inserted into the chamber before the test. After placing the mice in the environmentally modified chamber, the mouse was allowed to adapt to the environment for 4.0 min before being shown the light cue, which persisted for 2.0 minutes. The time spent freezing, as defined by an absence of movement (<200 pixels, >2.0 s), was recorded by a built-in infrared video camera. The time spent freezing during the 2.0 min immediately before and after presentation of the light cue was averaged using pre-installed imaging software (Ohara Medical Industry).

Visual acuity was measured 2 weeks after the AAV injection by observing the optokinetic responses of mice to rotating sinusoidal gratings presented on monitors (average luminance: 62 cd m$^{-2}$) surrounding the mouse (Optomotry, Cerebral Mechanics, Lethbridge, Canada)[16]. This test yields independent measures of right and left eye acuity based on the unequal sensitivities to pattern rotation direction, as the motion in the temporal-to-nasal direction dominates the tracking response[33]. Visual acuity data used in this study represented the averages of four trials conducted on four consecutive days. The results obtained by testing without using a mouse served as the negative control.

**In vitro on-target assessment**. Neuro2a genomic DNA was extracted 72 h post-transfection using a DNA extraction kit (QIAamp DNA Mini kit). PCR products were sub-cloned into T-vector (pTAC2; BioDynamics, Tokyo, Japan), which was used to transform a DH5a-competent cell (Toyobo, Osaka, Japan). DNA from single colonies (>50 clones) were amplified by colony direct PCR. Each PCR fragment was sequenced following a standard procedure using an ABI3130 genetic analyzer (Thermo Fisher Scientific)[34].

For preparation of DNA samples used for in vitro analysis by whole-genome sequencing, successfully transfected Neuro2a cells were used. In brief, Neuro2a cells were transfected with an SaCas9-2A-EGFP expression plasmid described above using lipofectamine 3000 transfection reagent (Thermo Fisher Scientific). After 72 h, EGFP-positive cells were selected using FASC aria II cell sorter (BD Biosciences, Franklin lakes, NJ), and genomic DNA were extracted from these cells using DNA extraction kit (QIAamp DNA Mini kit).

**In vivo on-target and off-target assessment**. The on-target site and the 14 off-target sites (listed in Supplementary Fig. 3 and Supplementary Table 3, assembly: GRCm38/mm10) predicted by CRISPOR (http://crispor.tefor.net/) were amplified with PCR using the primers listed in Supplementary Table 2. PCR products were subjected either to a T7E1 assay or Sanger sequencing of the PCR clones. We conducted a T7E1 assay for the 14 off-target sites, as described above in detail. PCR products of the on-target site and the 14 off-target sites were sub-cloned into T-vector (pTAC2; BioDynamics, Tokyo, Japan), which was used to transform a DH5a-competent cell (Toyobo, Osaka, Japan). DNA from single colonies (>50 clones for the on-target site and >50 clones each for the off-target site) were amplified by colony direct PCR. Each on-target and/or off-target PCR fragment was sequenced following a standard procedure using an ABI3130 genetic analyzer (Thermo Fisher Scientific)[34]. Classifications of the sequenced clones are as follows: Success, mutation replaced as planned; Cleavage site indel, insertion or deletion in either gRNA cleavage site without replacement of *IRD2* mutation; AAV integration, unplanned insertion of AAV genome fragment; Deletion, simple excision of the IRD2 mutation at the two gRNA cleavage sites without an insertion; Other indel, mutations that do not belong to any of the classifications above.

**Whole-genome sequencing and assessment of off-target sites**. For the genomic DNAs extracted from Neuro2A and mice, we performed whole-genome sequencing using the NovaSeq 6000 (Illumina, San Diego, CA, USA) sequencer with 151 bp paired-end reads. The amount of data per sample was made to exceed at least 100G bases. The sequencing library was constructed using the TruSeq Nano DNA Library Prep Kit (Illumina) according to the manufacturer's instructions. We prepared two reference genomes, mouse reference genome (mm10) and mm10 plus AAV genome (mm10 + AAV). The sequencing reads were separately aligned to mm10 and mm10 + AAV using BWA-mem (ver.0.7.17). Then, PCR duplicate reads were marked using Picard tools (ver.2.17.8). Base quality scores were recalibrated using GATK (ver.4.1.2.0) according to the GATK Best Practices (https://software.broadinstitute.org/gatk/best-practices/).

Single nucleotide variants (SNVs) and short insertions and deletions (indels) calling were performed for the WGS data to assess off-target sites. To detect variants with low variant allele frequency (VAF), we used the GATK4 Mutect2 software, which is used for somatic variant calling. The variants were

called according to the GATK Best Practices. Untransfected cells and untreated mouse data were used as normal control samples data in the Mutect2 variant calling.

In addition, to investigate whether integrations of the AAV occurred in the mice genome, we identified sequencing reads that were partially mapped (soft-clipped) to the AAV genome. From those sequencing reads, we extracted the subset of reads that were also partially mapped to the mouse genome (mm10). The extracted reads were evaluated for the presence of regions in which the AAV genome was inserted into the mouse genome.

**Determination of editing efficiency**. Absolute editing efficiency among rods were estimated by dividing *Gnat1* expression (RT-PCR) in the treated retinas of *Pde6c*$^{cpfl1/cpfl1}$*Gnat1*$^{IRD2/IRD2}$ mice by that in the retinas of untreated *Pde6c*$^{cpfl1/cpfl1}$ mice born with wildtype copies of *Gnat1* ($N = 4$). Similarly, the efficiency was estimated by dividing the 6 Hz ERG response amplitudes at $-1.0$ log cd s m$^{-2}$ in the treated eyes of *Pde6c*$^{cpfl1/cpfl1}$*Gnat1*$^{IRD2/IRD2}$ mice by those in the untreated eyes of *Pde6c*$^{cpfl1/cpfl1}$ mice (41.5 μV, average of four mice). When estimating the absolute efficiency by sequencing analysis of on-target site in an in vivo experiment, we corrected for the difference in detection efficiency (described below), arising from the difference in PCR amplicon size of the on-target site with an assumption that the difference in efficiency remains constant across various mixture of edited and unedited alleles. The proportion of rod photoreceptors among retinal cells were considered to be 0.75[19], which were also used to calculate genome editing efficacy among rods.

To determine the difference in detection efficiency of genome-edited success allele (670 bp amplicon) and unedited mutant *IRD2* allele (611 bp amplicon), 1:1 (50%) mixture (molecular ratio) of these alleles were PCR amplified, subcloned, and re-amplified by colony direct PCR in the same way as described in in vivo on-target and off-target assessment. The identity of the clones ($N = 53$) were determined by difference in the band size in agarose-gel electrophoresis. Against the expected sequence results of 26.5:26.5 clones, 16:37 clones were observed for success:mutant, indicating under-representation of the former by a factor of 16/26.5 = 0.60. This factor was 0.52 when competition between Success and even smaller Deletion (524 bp amplicon; the major editing outcome) was compared with a similar experiment (16:45 clones for success:deletion). In order to correct Success rate for unedited mutant *IRD2*, which comprised the major population of the clones analyzed, we carried out a similar experiment to 1:19 (5%), 1:9 (10%), 1:4 (20%), and 1:1 (50%) mixture of Success and unedited mutant *IRD2* allele (molecular ratio) followed by linear regression analysis (intercept -0.154, slope 0.528). In this case, the identity of the clones were determined by DNA sequencing. Using the results of regression analysis, we corrected only the rates of Success and unedited mutant *IRD2* allele. For example, observed absolute Success for MMEJ at 1 month was 0.047 (4.7%) then absolute corrected Success rate would be (4.7 + 0.154)/0.528 = ~9.185%. The calculation yields an underestimate of genome editing, as the Success represent the largest PCR amplicon, thus least efficiently detected, of all the other edited genomes.

**Statistical analysis**. Differences between pairs of groups were assessed with the paired Student's *t*-test (two-sided) for paired data and unpaired Student's *t*-test (two-sided) for other data. Differences between sets of three groups were assessed with an analysis of variance (ANOVA), followed by Tukey's test as a post hoc analysis. Linear regression analysis was carried out to generate Fig. 2d. All statistical analysis was performed with JMP (SAS Institute, Cary, NC). All values are expressed as the mean ± SEM. $P < 0.05$ was considered statistically significant.

**Reporting summary**. Further information on research design is available in the Nature Research Reporting Summary linked to this article.

## Data availability
The source data underlying Figs. 1d–h, 2b–h, 3a–d, 4b, c and Supplementary Figs. 1d, f, 2c, 5a, 6b, c and 7a–d are provided as a Source Data file. The datasets generated during and/or analyzed during the current study not listed in the Source Data file are available from the corresponding author on reasonable request.

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

## Acknowledgements

We thank Ms Misane Uchiike for her help with the experiments and Professor Carlo Rivolta for the critical comments on the manuscript. This work was supported in part by the Japan Agency for Medical Research and Development (KMN, 18ek0109213h0002). The manuscript was edited by a professional English editing service (Mr. Tim Hilts). We thank the Biomedical Research Core of Tohoku University Graduate School of Medicine for technical support.

## Author contributions

K.M.N. conceived and designed the experiments. K.M.N. and K.F. performed the experiments and analyzed the data. F.M. carried out in slico analysis. S.K. helped in vitro experiments. K.M.N. wrote the paper and K.M.N. and T.N. obtained the funding.

## Competing interests

K.M.N., K.F. and T.N. are listed as inventors in a patent application related to this work. The other authors declare no competing interests. The Departments of Advanced Ophthalmic Medicine and Ophthalmic Imaging and Information Analytics are endowed departments, supported by an unrestricted grant from Senju Pharmaceutical Co. (Osaka, Japan) and Topcon Co. Ltd. (Tokyo, Japan), respectively. These funders had no role in the study design, data collection and analysis, decision to publish, or preparation of the paper.
