## [Peer Review File · Nature Communications]

Reviewers' comments:

Reviewer #2 (Remarks to the Author):

The authors have addressed my queries and the new figure explaining the editing strategy in detail has made the manuscript easier to follow.

Reviewer #3 (Remarks to the Author):

The new version of the paper brings new important data allowing to better evaluate the work, but some sections are difficult to follow. Although the physiological studies appear clearer, several method descriptions remain problematic to really understand what was measured and interpreted to assess the efficacy of the gene editing process. In addition a discrepancy between the gene editing efficacy and protein expression is still present.

Major:

The authors claim that the gene editing efficacy is around 10% whereas the Figure 1 shows very few cells positive for GNAT1 (around 2%?). They explain in their rebuttal letter "it is not easy to confidently detect all the GNAT1-positive cells with retinal sections because of the significant background staining of...". This argument is not evidenced with the positive control labelling nor with non-treated animals for which the no background appears. Moreover in the Figure S6, it is clear that high expression of Gnat1 driven by a gene augmentation strategy resulted in a marked and broad increase of the GNAT1 expression, which is not the case in the gene editing result. In consequence, the gene editing efficacy appears much lower than 10%. The abstract has thus to be corrected in consequence (see also the comment below).

Reading this new version, I am not fully clear about the quantification method of the genome editing efficacy. Indeed, the authors claim that they have around 11% of success to correct the gene mutation. At a glance, I thought that this efficacy was estimated for all the photoreceptors (those with edited genome and the others where no editing was observed). But regarding the primers used, it seems that the authors analyzed only the region where an editing occurred. Indeed after BLASTING verification, one primer codes for GNAT1 KO mouse genome, whereas, for the other, it is difficult to identify its site. I guess (in view of the BLAST result) it is a sequence of the vector template. Thus, the 11% of efficacy is for correct editing for edited analyzed cells, not for all the photoreceptor population. This markedly reduces the interest of this study and may explain the low level of GNAT1 positive cells (see Fig. 1 and 2). In addition, some graphs of the Figure 2 are unclear. Fig. 2b is not explained and we don't know how this calculation was made. Do the authors take into account all the editing events and compared them with all retina genomes and make the percentage? If it is the case, this means that the editing efficacy is of about 2%. In addition, the results of the NoMHA group are surprising: the authors observed more editing events with a construction without homology arms in comparison to the therapeutic vector. What is their hypothesis? The calculation for Fig 2c and 2d needs to be better explained. A schema explaining the PCR strategy to analyze the gene editing efficacy (in supplementary material) would be welcome to better conduct the reader.

The title suggests that the gene editing is really efficient and can reestablish a robust visual function. I would prefer to see a title explaining that few cells corrected by gene editing can lead to a marked improvement of certain visual functions.

Minors:

Thank you for providing the results of the gene augmentation experiment which clearly reveal that

this strategy is much more efficient than the gene editing approach. Indeed, the scotopic threshold is 10⁻³ in OE and 1 in gene editing. One can observe a 4 log difference for the ERG results and for the PEV, the Fig 3a suggests a one log difference for a better sensitivity with the gene transfer (-5 versus -4). In consequence, the term similar in page 18 is not adequate.

Page 4, the authors wrote : " Furthermore, histology showed no sign of accelerated cone degeneration (Fig. 1d and S1c)." This sentence suggests that the gene editing approach has no effect on the cone degeneration. However, no evidences are provided to show that the gene editing occurred in the cones, but this is not the purpose of this study and the sentence has to be adequately changed.

In the methods, the authors did not mention whether the visual acuity test was performed in scotopic condition (after dark adaptation) or not.

Reviewer #4 (Remarks to the Author):

Overall the revised work appears robust and is certainly interesting, with very solid pre-clinical information which will help progress CRISPR/Cas to the clinic.

Clearly CRISPR/Cas9 MMEJ-based correction may in time be proven to be surpassed by "Prime Editing," though this is still likely to be sometime away.

I have no major suggestions for improvement for the manuscript.

Alex Hewitt

Reviewer's comments

Reviewer #2 (Remarks to the Author):

"The authors have addressed my queries and the new figure explaining the editing strategy in detail has made the manuscript easier to follow."

RESPONSE:

We are very pleased to know that the reviewer has been satisfied. We thank the reviewer for providing us a chance to greatly improve our manuscript.

Reviewer #3 (Remarks to the Author):

"The new version of the paper brings new important data allowing to better evaluate the work, but some sections are difficult to follow. Although the physiological studies appear clearer, several method descriptions remain problematic to really understand what was measured and interpreted to assess the efficacy of the gene editing process. In addition a

discrepancy between the gene editing efficacy and protein expression is still present.”

Response:

We are grateful for taking time to critically review our manuscript. We have tried our best to explain/ present data to answer the questions by the reviewer. This process has further reinforced the preexisting data, which we owe to the reviewer.

Major:

Comment 1:

“The authors claim that the gene editing efficacy is around 10% whereas the Figure 1 shows very few cells positive for GNAT1 (around 2%?). They explain in their rebuttal letter “it is not easy to confidently detect all the GNAT1-positive cells with retinal sections because of the significant background staining of...”. This argument is not evidenced with the positive control labelling nor with non-treated animals for which the no background appears. Moreover in the Figure S6, it is clear that high expression of Gnat1 driven by a gene augmentation strategy resulted in a marked and broad increase of the GNAT1 expression, which is not the case in the gene editing result. In consequence, the gene editing efficacy appears much lower than 10%. The abstract has thus to be corrected in consequence (see also the comment below).”

Response:

As the reviewer pointed out, genome editing efficacy visible at the protein level as tagged by immunohistochemistry in the previous submission differed from that estimated at genome, mRNA, and electrophysiological levels, which all pointed to an efficacy of ~10%.

Because the background staining and GNAT1 staining were both observed at the photoreceptor outer segments, we initially struggled a lot to distinguish the two. However, since the last revision, we have greatly improved our imaging strategy such that we now show better immunohistology images of also the eye treated with MMEJ-genome editing showing around 11% GNAT-positive outer segments relative to DAPI positive photoreceptors in histological sections (29 GNAT1-positive outer segments/ 256 DAPI positive nuclei). Unfortunately, although a larger number of GNAT1-positive outer segments can be observed also in the new retinal flatmount image, they are less clearly distinguishable and difficult to quantify. Nevertheless, the immunohistochemistry images newly provided are now more consistent with all other tests.

We apologize for confusing the reviewer by showing images of little background staining particularly for the flatmounts (middle panels in Figure 1b) in the previous submission. We have replaced these images to show more representative ones.

Comment 2

“Reading this new version, I am not fully clear about the quantification method of the genome editing efficacy. Indeed, the authors claim that they have around 11% of success to correct the gene mutation. At a glance, I thought that this efficacy was estimated for all the photoreceptors (those with edited genome and the others where no editing was observed). But regarding the primers used, it seems that the authors analyzed only the region where an editing occurred. Indeed after BLASTING verification, one primer codes for GNAT1 KO mouse genome, whereas, for the other it is difficult to identify its site. I guess (in view of the BLAST result) it is a sequence of the vector template. Thus, the 11% of efficacy is for correct editing for edited analyzed cells, not for all the photoreceptor population. This markedly

reduces the interest of this study and may explain the low level of GNAT1 positive cells (see Fig. 1 and 2)."

"In addition, some graphs of the Figure 2 are unclear. Fig. 2b is not explained and we don't know how this calculation was made. Do the authors take into account all the editing events and compared them with all retina genomes and make the percentage? If it is the case, this means that the editing efficacy is of about 2%."

"The calculation for Fig 2c and 2d needs to be better explained. A schema explaining the PCR strategy to analyze the gene editing efficacy (in supplementary material) would be welcome to better conduct the reader."

Response:

We thank the reviewer for highlighting the confusing calculation of genome editing efficiency. And we apologize for not being able to provide sufficient explanation. We have added explanation of the figure in addition to new experimental data to generate a better estimation of genome editing efficacy.

First of all, we provided illustration of primer location in Figure S4 and amended Table S2 to promote the understanding of the primer, which were designed on the mouse genome flanking the *IRD2* mutation.

As for Figure 2b the reviewer is right that all the editing events were considered and they were compared with all retina genomes. Thus for example, uncorrected absolute editing efficiency for MMEJ at 1 month can be calculated to be $0.111 \times 0.316 \times 1/0.75 = \sim 0.047$ (0.111: rate of "Success" among edited clones, 0.316: rate of total edited clones among all clones, 0.75: estimated rate of rod genome among total retinal genome) .

After MMEJ-mediated genome editing, PCR amplicons of ON-target site can be mostly

classified into three different sizes. The smallest size is the deletion of the genome between the two gRNA target sites followed by NHEJ (~524

bp). The largest is the genome successfully treated by genome editing which results in a 59 bp insertion (670 bp). The middle size (~611 bp) includes unedited “mutant” *IRD2* allele which is by far most abundant allele. We confirmed that detection efficiency of PCR/cloning used to generate data presented in Figure 2A is different between the ON-target sites of three different sizes, the successfully treated site being the least efficient and the most under-represented and the deletion being the most efficient and over-estimated. To partially correct for the problem, we determined the relationship between the actual and the observed rate of successfully treated genome (“Success”) and unedited “mutant” *IRD2* allele (by far the most frequent clone) by assuming linear regression (Figure 2d). We use this analysis (intercept -0.154, slope 0.528) to correct only the discrepancy between the “Success” and unedited “mutant” *IRD2* allele (Figure 2e). We have provided the detail of the methods at P27-29, lines 479-504, which reads;

“When estimating the absolute efficiency by sequencing analysis of on-target site in an in vivo experiment, we corrected for the difference in detection efficiency (described below), arising from the difference in PCR amplicon size of the on-target site with an assumption that the difference in efficiency remains constant across various mixture of edited and unedited alleles. The proportion of rod photoreceptors among retinal cells were considered to be 0.7519, which were also used to calculate genome editing efficacy among rods.

*To determine the difference in detection efficiency of genome edited “success” allele (670 bp amplicon) and unedited “mutant” *IRD2* allele (611 bp amplicon), 1:1 (50%) mixture (molecular ratio) of these alleles were PCR amplified, subcloned, and re-amplified by colony direct PCR in the same way as described in in vivo on-target and off-target assessment. The*

identity of the clones (N = 53) were determined by difference in the band size in agarose-gel electrophoresis. Against the expected sequence results of 26.5:26.5 clones, 16:37 clones were observed for success:mutant, indicating under-representation of the former by a factor of $16/26.5 = 0.60$. This factor was 0.52 when competition between “Success” and even smaller “Deletion” (524 bp amplicon; the major editing outcome) was compared with a similar experiment (16:45 clones for success:deletion). In order to correct “Success” rate for unedited “mutant” IRD2, which comprised the major population of the clones analyzed, we carried out the same experiment to 1:19 (5%), 1:9 (10%), 1:4 (20%), and 1:1 (50%) mixture of “Success” and unedited “mutant” IRD2 allele (molecular ratio) followed by linear regression analysis (intercept -0.154, slope 0.528). Using the results of regression analysis, we corrected only the rates of “Success” and unedited “mutant” IRD2 allele. For example, observed absolute “Success” for MMEJ at 1 month was 0.047 (4.7%) then absolute corrected “Success” rate would be $(4.7 + 0.154)/0.528 = \sim 9.185\%$. The calculation yields an underestimate of genome editing, as the “Success” represent the largest PCR amplicon, thus least efficiently detected, of all the other edited genomes.”

Comment 3

“In addition, the results of the NoMHA group are surprising: the authors observed more editing events with a construction without homology arms in comparison to the therapeutic vector. What is their hypothesis?”

Response:

Thank you for pointing this out. Because HITI and noMHA strategies share similar vector designs, it is not surprising that these two have similar editing rate. However, the breakup

could be largely different since, unlike NoMHA, inserted donor cannot be re-excised in HITI skewing the outcome toward decreased “Deletion”.

Meanwhile, results for MMEJ are consistent with the absolute editing efficacy measured by RT-PCR and ERG. As all the vectors except MMEJ vectors are expected to almost exclusively use NHEJ, we think that the potentially less efficient genome editing in MMEJ vectors could be accounted for by the competition between less efficient MMEJ and NHEJ (Xiong et al, Nucleic acids research 2015), if the difference observed is significant. On the other hand, an error in the calculation of genome editing results for NoTS has been corrected. As a result, *in vivo* editing rates are now more consistent with the *in vitro* data (Figure S4).

Xiong, X., Du, Z., Wang, Y., Feng, Z., Fan, P., Yan, C., ... & Zhang, J. (2015). 53BP1 promotes microhomology-mediated end-joining in G1-phase cells. Nucleic acids research, 43(3), 1659-1670.

Comment 4

“The title suggests that the gene editing is really efficient and can reestablish a robust visual function. I would prefer to see a title explaining that few cells corrected by gene editing can lead to a marked improvement of certain visual functions.”

Response:

It is true the majority of cells are not treated. Therefore, we have changed the title as follows;

“Single AAV-mediated mutation replacement genome editing in limited number of

photoreceptors mediate marked visual restoration”

Because the word limit for the title is 15, we regret that it was impossible to integrate all the requests without leaving out more important messages. I hope this is acceptable.

Minors:

Comment 5

“Thank you for providing the results of the gene augmentation experiment which clearly reveal that this strategy is much more efficient than the gene editing approach. Indeed, the scotopic threshold is 10⁻³ in OE and 1 in gene editing. One can observe a 4 log difference for the ERG results and for the PEV, the Fig 3a suggests a one log difference for a better sensitivity with the gene transfer (-5 versus -4). In consequence, the term similar in page 18 is not adequate.”

Response:

We are sorry that we did not provide enough details of the visual function tests, which appears to have led to a misunderstanding. Both light sensitivity and visual acuity cannot be simply compared on the basis of amplitudes of mass light response of the visual pathway, which is largely influenced by the total number of functional neurons, which is clearly greater following gene supplementation as shown in histology. Instead, these visual parameters are about the best achievable function by a defined group of successfully treated cells. For example, if you look at the data for -5.0 log.cd.s m⁻² in both P1-N1 and N1-P2 dose-response curve (Figure 3a), the responses are larger for the gene supplementation

(OE)-treated mice, which may or may not mean that they have better light sensitivity. However, responses at $-6.0 \log.cd.s m^{-2}$ becomes rapidly non-detectable similarly for both MMEJ- and gene supplementation (OE)-treated mice, consistent with the response threshold being rather similar between the two. When light sensitivity threshold was determined as defined by the dimmest light stimulus that can generate response larger than noise level (25 μV), there were no difference between the two methodologies (Figure 3a right lower panel), indicating that light sensitivity mediated by both treatments are indeed not very different. As for the ERG, the responses are directly proportional to the functional retinal neurons and are much less sensitive compared to flash VEP in detecting light sensitivity. Therefore, the ERG results in Figure S6 may give a false impression that the sensitivity of the retina is by far better following gene supplementation (OE) over MME, when in fact it only indicates that the functional photoreceptors are greater in number following the former treatment. That is why psychophysical tests probing cortical light response are used to assess light sensitivity in clinical practice and never ERG. From the beginning, we feared that the ERG data would confuse the readers. And that is exactly why we did not include the data in the initial submission although we had the data.

We have briefly explained this in the Discussion to guide the readers interpret the data correctly at P11, lines 182-189, which reads;

“The results showed that the gene supplementation can treat by far a larger number of retinal neurons compared to the mutation replacement genome editing, resulting in substantially larger ERG responses directly proportional to the increased number of light-responsive photoreceptors in the former. However, the light sensitivity as defined by dimmest recognizable light stimulus and visual acuity was not very different between the two treatment approaches (Figure 3a right lower panel). This is because thresholds of these

visual perceptions reflect functional integrity of defined number of photoreceptors rather the total number of treated retinal neurons.”

Comment 6

“Page 4, the authors wrote :” Furthermore, histology showed no sign of accelerated cone degeneration (Fig. 1d and S1c).” This sentence suggests that the gene editing approach has no effect on the cone degeneration. However, no evidences are provided to show that the gene editing occurred in the cones, but this is not the purpose of this study and the sentence has to be adequately changed.”

Response:

Reviewer is correct in that we show no evidence of genome editing in cones. Taking the reviewer’s comment into consideration, the sentence has been changed to;

“Furthermore, histology showed no sign of accelerated cone degeneration as a side effect of the treatment, although we have no evidence that genome editing occurs in cones (Fig. 1d and S1c).”

As has been shown in Figure S1, the *GRK1* promoter used in our vector is active also in the cones. Therefore, it is possible that the genome is edited also in these cells. In response to the request by Reviewer 2 in the last review who asked us to “*looked at the cones in any way to see if they have been affected by off target effects*” in the first review, we quantified the residual cones to show that there is no serious off-target effect leading to cell death, if genome editing took place.

Comment 7

“In the methods, the authors did not mention whether the visual acuity test was performed in scotopic condition (after dark adaptation) or not.”

Response:

For visual acuity tests, mice were not dark-adapted. As newly outlined in the Methods, the test monitor was set at 62 cd/m^{-2} and for Optokinetic response and 81 cd/m^{-2} and for pattern VEP, both in the lower range of monitor setting used in a regular work environment for humans. However, in the nearly completely blind mouse model treated, one does not need to dark-adapt or light-adapt to distinguish functions of the rods and the cones, respectively, as measurable visual function should be mediated almost exclusively by the genome-edited functional rods expressing GNAT1.

Reviewer #4 (Remarks to the Author):

“Overall the revised work appears robust and is certainly interesting, with very solid pre-clinical information which will help progress CRISPR/Cas to the clinic.

Clearly CRISPR/Cas9 MMEJ-based correction may in time be proven to be surpassed by

“Prime Editing,” though this is still likely to be sometime away.

I have no major suggestions for improvement for the manuscript.”

RESPONSE:

Thank you very much for the encouraging and insightful comments. We are hoping to improve various aspects of the approach before we consider applying this approach to our

patients. In the meantime, we are very keen to trying out “Prime Editing” to see how it works in the retinal degeneration models.

REVIEWERS' COMMENTS:

Reviewer #3 (Remarks to the Author):

The authors did major efforts to respond to all concerns and to present new results and explanations. The manuscript is now comprehensive and supports the claims. Please, pay just attention to the legend of the Figure 3 for which, b and c were inverted. "b" necessitates also more descriptions for the 3 panels, only "Pattern VEPs. N = 11, and 10 for MMEJ and untreated, respectively." are mentioned.

REVIEWERS' COMMENTS

Reviewer #3 (Remarks to the Author):

COMMENT:

"The authors did major efforts to respond to all concerns and to present new results and explanations. The manuscript is now comprehensive and supports the claims. Please, pay just attention to the legend of the Figure 3 for which, b and c were inverted. "b" necessitates also more descriptions for the 3 panels, only "Pattern VEPs. N = 11, and 10 for MMEJ and untreated, respectively." are mentioned."

ANSWER:

We are grateful to the reviewer for the enormous effort in carefully going over our work and providing insightful comments and spotting grave mistakes, which have helped us to substantially improved the manuscript. We have now fixed the labeling of the Figure so that it matches the main text and the Figure legend. We have also re-labeled the Data source file accordingly.